# Hierarchical Reinforcement Learning for Zero-shot Generalization with Subtask Dependencies

**Sungryull Sohn**
University of Michigan
srsohn@umich.edu

**Junhyuk Oh**[*]
University of Michigan
junhyuk@google.com

**Honglak Lee**
Google Brain
University of Michigan
honglak@google.com

## Abstract

We introduce a new RL problem where the agent is required to generalize to a previously-unseen environment characterized by a subtask graph which describes a set of subtasks and their dependencies. Unlike existing hierarchical multitask RL approaches that explicitly describe what the agent should do at a high level, our problem only describes properties of subtasks and relationships among them, which requires the agent to perform complex reasoning to find the optimal subtask to execute. To solve this problem, we propose a *neural subtask graph solver* (NSGS) which encodes the subtask graph using a recursive neural network embedding. To overcome the difficulty of training, we propose a novel non-parametric gradient-based policy, *graph reward propagation*, to pre-train our NSGS agent and further finetune it through actor-critic method. The experimental results on two 2D visual domains show that our agent can perform complex reasoning to find a near-optimal way of executing the subtask graph and generalize well to the unseen subtask graphs. In addition, we compare our agent with a Monte-Carlo tree search (MCTS) method showing that our method is much more efficient than MCTS, and the performance of NSGS can be further improved by combining it with MCTS.

## 1 Introduction

Developing the ability to execute many different tasks depending on given task descriptions and generalize over unseen task descriptions is an important problem for building scalable reinforcement learning (RL) agents. Recently, there have been a few attempts to define and solve different forms of task descriptions such as natural language [1, 2] or formal language [3, 4]. However, most of the prior works have focused on task descriptions which explicitly specify what the agent should do at a high level, which may not be readily available in real-world applications.

To further motivate the problem, let's consider a scenario in which an agent needs to generalize to a complex novel task by performing a composition of subtasks where the task description and dependencies among subtasks may change depending on the situation. For example, a human user could ask a physical household robot to make a meal in an hour. A meal may be served with different combinations of dishes, each of which takes a different amount of cost (e.g., time) and gives a different amount of reward (e.g., user satisfaction) depending on the user preferences. In addition, there can be complex dependencies between subtasks. For example, a bread should be sliced before toasted, or an omelette and an egg sandwich cannot be made together if there is only one egg left. Due to such complex dependencies as well as different rewards and costs, it is often cumbersome for human users to manually provide the optimal sequence of subtasks (e.g., "fry an egg and toast a bread"). Instead, the agent should learn to act in the environment by figuring out the optimal sequence of subtasks that gives the maximum reward within a time budget just from properties and dependencies of subtasks.

The goal of this paper is to formulate and solve such a problem, which we call *subtask graph execution*, where the agent should execute the given *subtask graph* in an optimal way as illustrated in Figure 1.

---

[*]Now at DeepMind.

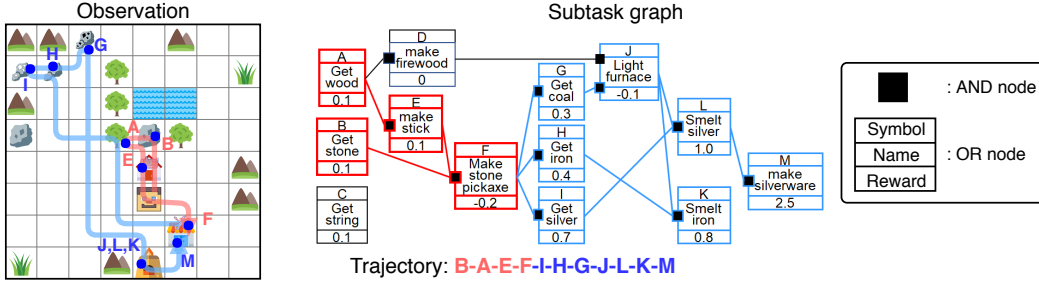

Figure 1: Example task and our agent's trajectory. The agent is required to execute subtasks in the optimal order to maximize the reward within a time limit. The subtask graph describes subtasks with the corresponding rewards (e.g., subtask L gives 1.0 reward) and dependencies between subtasks through AND and OR nodes. For instance, the agent should first get the firewood (D) OR coal (G) to light a furnace (J). In this example, our agent learned to execute subtask F and its preconditions (shown in red) as soon as possible, since it is a precondition of many subtasks even though it gives a negative reward. After that, the agent mines minerals that require stone pickaxe and craft items (shown in blue) to achieve a high reward.

A subtask graph consists of subtasks, corresponding rewards, and dependencies among subtasks in logical expression form where it subsumes many existing forms (e.g., sequential instructions [1]). This allows us to define many complex tasks in a principled way and train the agent to find the optimal way of executing such tasks. Moreover, we aim to solve the problem without explicit search or simulations so that our method can be more easily applicable to practical real-world scenarios, where real-time performance (i.e., fast decision-making) is required and building the simulation model is extremely challenging.

To solve the problem, we propose a new deep RL architecture, called *neural subtask graph solver* (NSGS), which encodes a subtask graph using a recursive-reverse-recursive neural network (R3NN) [5] to consider the long-term effect of each subtask. Still, finding the optimal sequence of subtasks by reflecting the long-term dependencies between subtasks and the context of observation is computationally intractable. Therefore, we found that it is extremely challenging to learn a good policy when it's trained from scratch. To address the difficulty of learning, we propose to pre-train the NSGS to approximate our novel non-parametric policy called *graph reward propagation policy*. The key idea of the graph reward propagation policy is to construct a differentiable representation of the subtask graph such that taking a gradient over the reward results in propagating reward information between related subtasks, which is used to find a reasonably good subtask to execute. After the pre-training, our NSGS architecture is finetuned using the actor-critic method.

The experimental results on 2D visual domains with diverse subtask graphs show that our agent implicitly performs complex reasoning by taking into account long-term subtask dependencies as well as the cost of executing each subtask from the observation, and it can successfully generalize to unseen and larger subtask graphs. Finally, we show that our method is computationally much more efficient than Monte-Carlo tree search (MCTS) algorithm, and the performance of our NSGS agent can be further improved by combining with MCTS, achieving a near-optimal performance.

Our contributions can be summarized as follows: (1) We propose a new challenging RL problem and domain with a richer and more general form of graph-based task descriptions compared to the recent works on multitask RL. (2) We propose a deep RL architecture that can execute arbitrary *unseen* subtask graphs and observations. (3) We demonstrate that our method outperforms the state-of-the-art search-based method (e.g., MCTS), which implies that our method can efficiently approximate the solution of an intractable search problem without performing any search. (4) We further show that our method can also be used to augment MCTS, which significantly improves the performance of MCTS with a much less amount of simulations.

## 2 Related Work

**Programmable Agent**   The idea of learning to execute a given program using RL was introduced by programmable hierarchies of abstract machines (PHAMs) [6–8]. PHAMs specify a partial policy using a set of hierarchical finite state machines, and the agent learns to execute the partial program. A different way of specifying a partial policy was explored in the deep RL framework [4]. Other approaches used a program as a form of task description rather than a partial policy in the context of multitask RL [1, 3]. Our work also aims to build a programmable agent in that we train the agent to execute a given task. However, most of the prior work assumes that the program specifies what to do,

and the agent just needs to learn how to do it. In contrast, our work explores a new form of program, called *subtask graph* (see Figure 1), which describes properties of subtasks and dependencies between them, and the agent is required to figure out what to do as well as how to do it.

**Hierarchical Reinforcement Learning**    Many hierarchical RL approaches have been proposed to solve complex decision problems via multiple levels of temporal abstractions [9–13]. Our work builds upon the prior work in that a high-level controller focuses on finding the optimal subtask, while a low-level controller focuses on executing the given subtask. In this work, we focus on how to train the high-level controller for generalizing to novel complex dependencies between subtasks.

**Classical Search-Based Planning**    One of the most closely related problems is the planning problem considered in hierarchical task network (HTN) approaches [14–18] in that HTNs also aim to find the optimal way to execute tasks given subtask dependencies. However, they aim to execute a single goal task, while the goal of our problem is to maximize the cumulative reward in RL context. Thus, the agent in our problem not only needs to consider dependencies among subtasks but also needs to infer the cost from the observation and deal with stochasticity of the environment. These additional challenges make it difficult to apply such classical planning methods to solve our problem.

**Motion Planning**    Another related problem to our subtask graph execution problem is motion planning (MP) problem [19–23]. MP problem is often mapped to a graph, and reduced to a graph search problem. However, different from our problem, the MP approaches aim to find an optimal path to the goal in the graph while avoiding obstacles similar to HTN approaches.

## 3    Problem Definition

### 3.1    Preliminary: Multitask Reinforcement Learning and Zero-Shot Generalization

We consider an agent presented with a task drawn from some distribution as in [4, 24]. We model each task as Markov Decision Process (MDP). Let $G \in \mathcal{G}$ be a task parameter available to agent drawn from a distribution $P(G)$ where $G$ defines the task and $\mathcal{G}$ is a set of all possible task parameters. The goal is to maximize the expected reward over the whole distribution of MDPs: $\int P(G)J(\pi, G)dG$, where $J(\pi, G) = \mathbb{E}_\pi[\sum_{t=0}^{T} \gamma^t r_t]$ is the expected return of the policy $\pi$ given a task defined by $G$, $\gamma$ is a discount factor, $\pi : \mathcal{S} \times \mathcal{G} \to \mathcal{A}$ is a multitask policy that we aim to learn, and $r_t$ is the reward at time step $t$. We consider a zero-shot generalization where only a subset of tasks $\mathcal{G}_{train} \subset \mathcal{G}$ is available to agent during training, and the agent is required to generalize over a set of unseen tasks $\mathcal{G}_{test} \subset \mathcal{G}$ for evaluation, where $\mathcal{G}_{test} \cap \mathcal{G}_{train} = \phi$.

### 3.2    Subtask Graph Execution Problem

The *subtask graph execution* problem is a multitask RL problem with a specific form of task parameter $G$ called *subtask graph*. Figure 1 illustrates an example subtask graph and environment. The task of our problem is to execute given $N$ subtasks in an optimal order to maximize reward within a time budget, where there are complex dependencies between subtasks defined by the subtask graph. We assume that the agent has learned a set of *options* ($\mathcal{O}$) [11, 25, 9] that performs subtasks by executing one or more primitive actions.

**Subtask Graph and Environment**    We define the terminologies as follows:

- **Precondition**: A *precondition* of subtask is defined as a logical expression of subtasks in sum-of-products (SoP) form where multiple `AND` terms are combined with an `OR` term (e.g., the precondition of subtask J in Figure 1 is `OR(AND(D), AND(G))`).
- **Eligibility vector**: $\mathbf{e}_t = [e_t^1, \dots, e_t^N]$ where $e_t^i = 1$ if subtask $i$ is *eligible* (i.e., the precondition of subtask is satisfied and it has never been executed by the agent) at time $t$, and 0 otherwise.
- **Completion vector**: $\mathbf{x}_t = [x_t^1, \dots, x_t^N]$ where $x_t^i = 1$ if subtask $i$ has been executed by the agent while it is eligible, and 0 otherwise.
- **Subtask reward vector**: $\mathbf{r} = [r^1, \dots, r^N]$ specifies the reward for executing each subtask.
- **Reward**: $r_t = r^i$ if the agent executes the subtask $i$ while it is eligible, and $r_t = 0$ otherwise.
- **Time budget**: $step_t \in \mathbb{R}$ is the remaining time-steps until episode termination.
- **Observation**: $\mathbf{obs}_t \in \mathbb{R}^{H \times W \times C}$ is a visual observation at time $t$ as illustrated in Figure 1.

To summarize, a subtask graph $G$ defines $N$ subtasks with corresponding rewards $\mathbf{r}$ and the preconditions. The state input at time $t$ consists of $\mathbf{s}_t = \{\mathbf{obs}_t, \mathbf{x}_t, \mathbf{e}_t, step_t\}$. The goal is to find a policy $\pi : \mathbf{s}_t, G \mapsto \mathbf{o}_t$ which maps the given context of the environment to an *option* ($\mathbf{o}_t \in \mathcal{O}$).

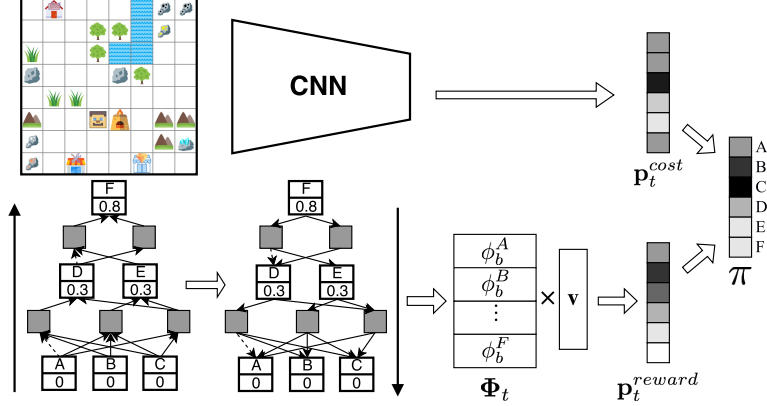

Figure 2: Neural subtask graph solver architecture. The task module encodes subtask graph through a bottom-up and top-down process, and outputs the reward score $\mathbf{p}_t^{reward}$. The observation module encodes observation using CNN and outputs the cost score $\mathbf{p}_t^{cost}$. The final policy is a softmax policy over the sum of two scores.

**Challenges**    Our problem is challenging due to the following aspects:

- **Generalization**: Only a subset of subtask graphs ($\mathcal{G}_{train}$) is available during training, but the agent is required to execute previously unseen and larger subtask graphs ($\mathcal{G}_{test}$).

- **Complex reasoning**: The agent needs to infer the long-term effect of executing individual subtasks in terms of reward and cost (e.g., time) and find the optimal sequence of subtasks to execute without any explicit supervision or simulation-based search. We note that it may not be easy even for humans to find the solution without explicit search due to the exponentially large solution space.

- **Stochasticity**: The outcome of subtask execution is stochastic in our setting (for example, some objects are randomly moving). Therefore, the agent needs to consider the expected outcome when deciding which subtask to execute.

## 4   Method

Our *neural subtask graph solver* (NSGS) is a neural network which consists of a *task module* and an *observation module* as shown in Figure 2. The task module encodes the precondition of each subtask via bottom-up process and propagates the information about future subtasks and rewards to preceding subtasks (i.e., pre-conditions) via the top-down process. The observation module learns the correspondence between a subtask and its target object, and the relation between the locations of objects in the observation and the time cost. However, due to the aforementioned challenge (i.e., *complex reasoning*) in Section 3.2, learning to execute the subtask graph only from the reward is extremely challenging. To facilitate the learning, we propose *graph reward propagation policy* (GRProp), a non-parametric policy that propagates the reward information between related subtasks to model their dependencies. Since our GRProp acts as a good initial policy, we train the NSGS to approximate the GRProp policy through policy distillation [26, 27], and finetune it through actor-critic method with generalized advantage estimation (GAE) [28] to maximize the reward. Section 4.1 describes the NSGS architecture, and Section 4.2 describes how to construct the GRProp policy.

### 4.1   Neural Subtask Graph Solver

**Task Module**    Given a subtask graph $G$, the remaining time steps $step_t \in \mathbb{R}$, an eligibility vector $\mathbf{e}_t$ and a completion vector $\mathbf{x}_t$, we compute a context embedding using recursive-reverse-recursive neural network (R3NN) [5] as follows:

$$\phi_{bot,o}^i = b_{\theta_o}\left(x_t^i, e_t^i, step_t, \sum_{j \in Child_i} \phi_{bot,a}^j\right), \qquad \phi_{bot,a}^j = b_{\theta_a}\left(\sum_{k \in Child_j}\left[\phi_{bot,o}^k, w_+^{j,k}\right]\right), \quad (1)$$

$$\phi_{top,o}^i = t_{\theta_o}\left(\phi_{bot,o}^i, r^i, \sum_{j \in Par_i}\left[\phi_{top,a}^j, w_+^{i,j}\right]\right), \quad \phi_{top,a}^j = t_{\theta_a}\left(\phi_{bot,a}^j, \sum_{k \in Par_j} \phi_{top,o}^k\right), \quad (2)$$

where $[\cdot]$ is a concatenation operator, $b_\theta, t_\theta$ are the bottom-up and top-down encoding function, $\phi_{bot,a}^i, \; \phi_{top,a}^i$ are the bottom-up and top-down embedding of $i$-th AND node respectively, and

$\phi^i_{bot,o}$, $\phi^i_{top,o}$ are the bottom-up and top-down embedding of $i$-th OR node respectively (see Appendix for the detail). The $w^{i,j}_+$, $Child_i$, and $Parent_i$ specifies the connections in the subtask graph $G$. Specifically, $w^{i,j}_+ = 1$ if $j$-th OR node and $i$-th AND node are connected without NOT operation, $-1$ if there is NOT connection and $0$ if not connected, and $Child_i, Parent_i$ represent a set of $i$-th node's children and parents respectively. The embeddings are transformed to reward scores via: $\mathbf{p}^{reward}_t = \mathbf{\Phi}^\top_{top}\mathbf{v}$, where $\mathbf{\Phi}_{top} = [\phi^1_{top,o}, \dots, \phi^N_{top,o}] \in \mathbb{R}^{E \times N}$, $E$ is the dimension of the top-down embedding of OR node, and $\mathbf{v} \in \mathbb{R}^E$ is a weight vector for reward scoring.

**Observation Module**   The observation module encodes the input observation $\mathbf{obs}_t$ using a convolutional neural network (CNN) and outputs a cost score:

$$\mathbf{p}^{cost}_t = \text{CNN}(\mathbf{obs}_t, step_t). \tag{3}$$

where $step_t$ is the number of remaining time steps. An ideal observation module would learn to estimate high score for a subtask if the target object is close to the agent because it would require less cost (i.e., time). Also, if the expected number of step required to execute a subtask is larger than the remaining step, ideal agent would assign low score. The NSGS policy is a softmax policy:

$$\pi(\mathbf{o}_t|\mathbf{s}_t, \mathbf{G}) = \text{Softmax}(\mathbf{p}^{reward}_t + \mathbf{p}^{cost}_t), \tag{4}$$

which adds reward scores and cost scores.

## 4.2   Graph Reward Propagation Policy: Pre-training Neural Subtask Graph Solver

Intuitively, the graph reward propagation policy is designed to put high probabilities over subtasks that are likely to maximize the sum of *modified and smoothed* reward $\widetilde{U}_t$ at time $t$, which will be defined in Eq. 9. Let $\mathbf{x}_t$ be a completion vector and $\mathbf{r}$ be a subtask reward vector (see Section 3 for definitions). Then, the sum of reward until time-step $t$ is given as:

$$U_t = \mathbf{r}^T \mathbf{x}_t. \tag{5}$$

We first modify the reward formulation such that it gives a half of subtask reward for satisfying the preconditions and the rest for executing the subtask to encourage the agent to satisfy the precondition of a subtask with a large reward:

$$\widehat{U}_t = \mathbf{r}^T(\mathbf{x}_t + \mathbf{e}_t)/2. \tag{6}$$

Let $y^j_{AND}$ be the output of $j$-th AND node. The eligibility vector $\mathbf{e}_t$ can be computed from the subtask graph $G$ and $\mathbf{x}_t$ as follows:

$$e^i_t = \underset{j \in Child_i}{\text{OR}}\left(y^j_{AND}\right), \quad y^j_{AND} = \underset{k \in Child_j}{\text{AND}}\left(\widehat{x}^{j,k}_t\right), \quad \widehat{x}^{j,k}_t = x^k_t w^{j,k} + (1 - x^k_t)(1 - w^{j,k}), \tag{7}$$

where $w^{j,k} = 0$ if there is a NOT connection between $j$-th node and $k$-th node, otherwise $w^{j,k} = 1$. Intuitively, $\widehat{x}^{j,k}_t = 1$ when $k$-th node does not violate the precondition of $j$-th node. Note that $\tilde{U}_t$ is not differentiable with respect to $\mathbf{x}_t$ because $\text{AND}(\cdot)$ and $\text{OR}(\cdot)$ are not differentiable. To derive our graph reward propagation policy, we propose to substitute $\text{AND}(\cdot)$ and $\text{OR}(\cdot)$ functions with "smoothed" functions $\widetilde{\text{AND}}$ and $\widetilde{\text{OR}}$ as follows:

$$\widetilde{e}^i_t = \underset{j \in Child_i}{\widetilde{\text{OR}}}\left(\widetilde{y}^j_{AND}\right), \quad \widetilde{y}^j_{AND} = \underset{k \in Child_j}{\widetilde{\text{AND}}}\left(\widehat{x}^{j,k}_t\right), \quad (8)$$

where $\widetilde{\text{AND}}$ and $\widetilde{\text{OR}}$ were implemented as scaled sigmoid and tanh functions as illustrated by Figure 3 (see Appendix for details). With the smoothed operations, the sum of smoothed and modified reward is given as:

$$\widetilde{U}_t = \mathbf{r}^T(\mathbf{x}_t + \widetilde{\mathbf{e}}_t)/2. \tag{9}$$

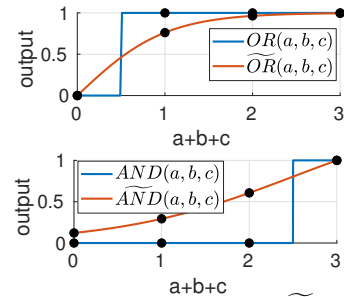

Figure 3: Visualization of OR, $\widetilde{\text{OR}}$, AND, and $\widetilde{\text{AND}}$ operations with three inputs (a,b,c). These smoothed functions are defined to handle arbitrary number of operands (see Appendix).

Finally, the graph reward propagation policy is a softmax policy,

$$\pi(\mathbf{o}_t|\mathbf{x}_t, G) = \text{Softmax}\left(\nabla_{\mathbf{x}_t}\widetilde{U}_t\right) = \text{Softmax}\left(\frac{1}{2}\mathbf{r}^T + \frac{1}{2}\mathbf{r}^T\nabla_{\mathbf{x}_t}\widetilde{\mathbf{e}}_t\right), \tag{10}$$

that is the softmax of the gradient of $\widetilde{U}_t$ with respect to $\mathbf{x}_t$.

## 4.3 Policy Optimization

The NSGS is first trained through policy distillation by minimizing the KL divergence between NSGS and teacher policy (GRProp) as follows:

$$\nabla_\theta \mathcal{L}_1 = \mathbb{E}_{G \sim \mathcal{G}_{train}} \left[ \mathbb{E}_{s \sim \pi_\theta^G} \left[ \nabla_\theta D_{KL} \left( \pi_T^G || \pi_\theta^G \right) \right] \right], \tag{11}$$

where $\theta$ is the parameter of NSGS, $\pi_\theta^G$ is the simplified notation of NSGS policy with subtask graph $G$, $\pi_T^G$ is the simplified notation of teacher (GRProp) policy with subtask graph $G$, $D_{KL}$ is KL divergence, and $\mathcal{G}_{train}$ is the training set of subtask graphs. After policy distillation, we finetune NSGS agent in an end-to-end manner using actor-critic method with GAE [28] as follows:

$$\nabla_\theta \mathcal{L}_2 = \mathbb{E}_{G \sim \mathcal{G}_{train}} \left[ \mathbb{E}_{s \sim \pi_\theta^G} \left[ -\nabla_\theta \log \pi_\theta^G \sum_{l=0}^{\infty} \left( \prod_{n=0}^{l-1} (\gamma\lambda)^{k_n} \right) \delta_{t+l} \right] \right], \tag{12}$$

$$\delta_t = r_t + \gamma^{k_t} V_{\theta'}^\pi(\mathbf{s}_{t+1}, G) - V_{\theta'}^\pi(\mathbf{s}_t, G), \tag{13}$$

where $k_t$ is the duration of option $\mathbf{o}_t$, $\gamma$ is a discount factor, $\lambda \in [0, 1]$ is a weight for balancing between bias and variance of the advantage estimation, and $V_{\theta'}^\pi$ is the critic network parameterized by $\theta'$. During training, we update the critic network to minimize $\mathbb{E}\left[ (R_t - V_{\theta'}^\pi(\mathbf{s}_t, G))^2 \right]$, where $R_t$ is the discounted cumulative reward at time $t$. The complete procedure for training our NSGS agent is summarized in Algorithm 1. We used $\eta_d$=1e-4, $\eta_c$=3e-6 for distillation and $\eta_{ac}$=1e-6, $\eta_c$=3e-7 for fine-tuning in the experiment.

---

**Algorithm 1** Policy optimization

---

1: **for** iteration $n$ **do**
2:      Sample $G \sim \mathcal{G}_{train}$
3:      $\mathcal{D} = \{(\mathbf{s}_t, \mathbf{o}_t, r_t, R_t, step_t), \ldots\} \sim \pi_\theta^G$                ▷ do rollout
4:      $\theta' \leftarrow \theta' + \eta_c \sum_{\mathcal{D}} \left( \nabla_{\theta'} V_{\theta'}^\pi(\mathbf{s}_t, G) \right) \left( R_t - V_{\theta'}^\pi(\mathbf{s}_t, G) \right)$      ▷ update critic
5:      **if** distillation **then**
6:          $\theta \leftarrow \theta + \eta_d \sum_{\mathcal{D}} \nabla_\theta D_{KL} \left( \pi_T^G || \pi_\theta^G \right)$             ▷ update policy
7:      **else if** fine-tuning **then**
8:          Compute $\delta_t$ from Eq. 13 for all $t$
9:          $\theta \leftarrow \theta + \eta_{ac} \sum_{\mathcal{D}} \nabla_\theta \log \pi_\theta^G \sum_{l=0}^{\infty} \left( \prod_{n=0}^{l-1} (\gamma\lambda)^{k_n} \right) \delta_{t+l}$      ▷ update policy

---

## 5 Experiment

In the experiment, we investigated the following research questions: 1) Does GRProp outperform other heuristic baselines (e.g., greedy policy, etc.)? 2) Can NSGS deal with complex subtask dependencies, delayed reward, and the stochasticity of the environment? 3) Can NSGS generalize to unseen subtask graphs? 4) How does NSGS perform compared to MCTS? 5) Can NSGS be used to improve MCTS?

### 5.1 Environment

We evaluated the performance of our agents on two domains: **Mining** and **Playground** that are developed based on MazeBase [29]. We used a pre-trained subtask executer for each domain. The episode length (time budget) was randomly set for each episode in a range such that GRProp agent executes $60\% - 80\%$ of subtasks on average. The subtasks in the higher layer in subtask graph are designed to give larger reward (see Appendix for details).

**Mining** domain is inspired by Minecraft (see Figures 1 and 5). The agent may pickup raw materials in the world, and use it to craft different items on different craft stations. There are two forms of preconditions: 1) an item may be an ingredient for building other items (e.g., stick and stone are ingredients of stone pickaxe), and 2) some tools are required to pick up some objects (e.g., agent need stone pickaxe to mine iron ore). The agent can use the item multiple times after picking it once. The set of subtasks and preconditions are hand-coded based on the crafting recipes in Minecraft, and used as a template to generate 640 random subtask graphs. We used 200 for training and 440 for testing.

**Playground** is a more flexible and challenging domain (see Figure 6). The subtask graph in Playground was randomly generated, hence its precondition can be any logical expression and the reward

may be delayed. Some of the objects randomly move, which makes the environment stochastic. The agent was trained on small subtask graphs, while evaluated on much larger subtask graphs (See Table 1). The set of subtasks is $\mathcal{O} = \mathcal{A}_{int} \times \mathcal{X}$, where $\mathcal{A}_{int}$ is a set of primitive actions to interact with objects, and $\mathcal{X}$ is a set of all types of interactive objects in the domain. We randomly generated 500 graphs for training and 2,000 graphs for testing. Note that the task in playground domain subsumes many other hierarchical RL domains such as Taxi [30], Minecraft [1] and XWORLD [2]. In addition, we added the following components into subtask graphs to make the task more challenging:

- Distractor subtask: A subtask with only NOT connection to parent nodes in the subtask graph. Executing this subtask may give an immediate reward, but it may make other subtasks ineligible.
- Delayed reward: Agent receives no reward from subtasks in the lower layers, but it should execute some of them to make higher-level subtasks eligible (see Appendix for fully-delayed reward case).

## 5.2 Agents

We evaluated the following policies:

- **Random** policy executes any eligible sub-task.
- **Greedy** policy executes the eligible subtask with the largest reward.
- **Optimal** policy is computed from exhaustive search on *eligible* subtasks.
- **GRProp** (Ours) is graph reward propagation policy.
- **NSGS** (Ours) is distilled from GRProp policy and finetuned with actor-critic.
- **Independent** is an LSTM-based baseline trained on each subtask graph independently, similar to Independent model in [4]. It takes the same set of input as NSGS except the subtask graph.

To our best knowledge, existing work on hierarchical RL cannot directly address our problem with a subtask graph input. Instead, we evaluated an instance of hierarchical RL method (**Independent** agent) in **adaptation** setting, as discussed in Section 5.3.

| Subtask Graph Setting | | | | | |
|---|---|---|---|---|---|
| | | Playground | | | Mining |
| Task | **D1** | **D2** | **D3** | **D4** | **Eval** |
| Depth | 4 | 4 | 5 | 6 | 4-10 |
| Subtask | 13 | 15 | 16 | 16 | 10-26 |

| Zero-Shot Performance | | | | | |
|---|---|---|---|---|---|
| | | Playground | | | Mining |
| Task | **D1** | **D2** | **D3** | **D4** | **Eval** |
| NSGS (Ours) | **.820** | **.785** | **.715** | **.527** | **8.19** |
| GRProp (Ours) | .721 | .682 | .623 | .424 | 6.16 |
| Greedy | .164 | .144 | .178 | .228 | 3.39 |
| Random | 0 | 0 | 0 | 0 | 2.79 |

| Adaptation Performance | | | | | |
|---|---|---|---|---|---|
| | | Playground | | | Mining |
| Task | **D1** | **D2** | **D3** | **D4** | **Eval** |
| NSGS (Ours) | **.828** | **.797** | **.733** | **.552** | **8.58** |
| Independent | .346 | .296 | .193 | .188 | 3.89 |

Table 1: Generalization performance on unseen and larger subtask graphs. (Playground) The subtask graphs in **D1** have the same graph structure as training set, but the graph was unseen. The subtask graphs in **D2**, **D3**, and **D4** have (unseen) larger graph structures. (Mining) The subtask graphs in **Eval** are unseen during training. NSGS outperforms other compared agents on all the task and domain.

## 5.3 Quantitative Result

**Training Performance** The learning curves of NSGS and performance of other agents are shown in Figure 4. Our GRProp policy significantly outperforms the Greedy policy. This implies that the proposed idea of back-propagating the reward gradient captures long-term dependencies among subtasks to some extent. We also found that NSGS further improves the performance through fine-tuning with actor-critic method. We hypothesize that NSGS learned to estimate the expected costs of executing subtasks from the observations and consider them along with subtask graphs.

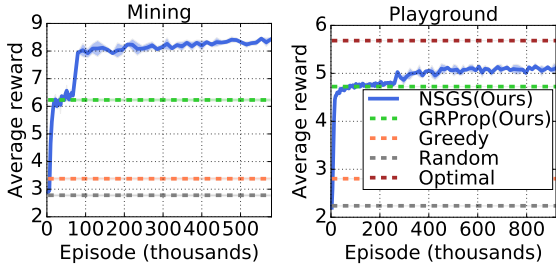

Figure 4: Learning curves on Mining and Playground domain. NSGS is distilled from GRProp on 77K and 256K episodes, respectively, and finetuned after that.

**Generalization Performance** We considered two different types of generalization: a **zero-shot** setting where agent must immediately achieve good performance on unseen subtask graphs without learning, and an **adaptation** setting where agent can learn about task through the interaction with environment. Note that Independent agent was evaluated in adaptation setting only since it has no ability to generalize as it does not take subtask graph as input. Particularly, we tested agents on larger subtask graphs by varying the number of layers of the subtask graphs from four to six with a larger

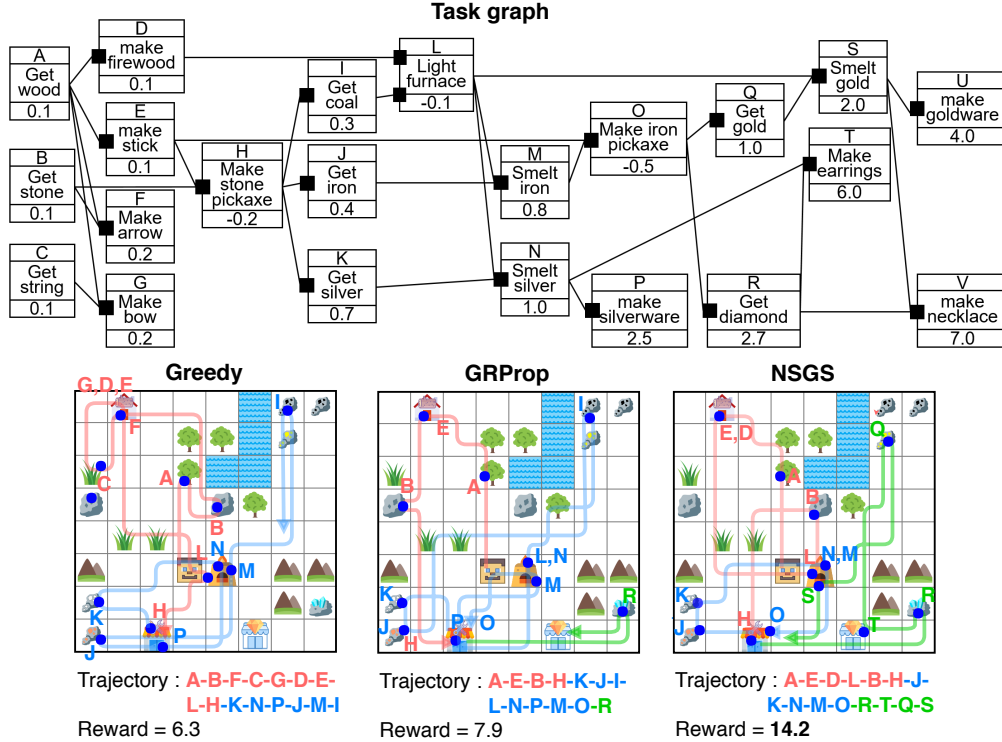

Figure 5: Example trajectories of Greedy, GRProp, and NSGS agents given 75 steps on Mining domain. We used different colors to indicate that agent has different types of pickaxes: red (no pickaxe), blue (stone pickaxe), and green (iron pickaxe). Greedy agent prefers subtasks C, D, F, and G to H and L since C, D, F, and G gives positive immediate reward, whereas NSGS and GRProp agents find a short path to make stone pickaxe, focusing on subtasks with higher long-term reward. Compared to GRProp, the NSGS agent can find a shorter path to make an iron pickaxe, and succeeds to execute more number of subtasks.

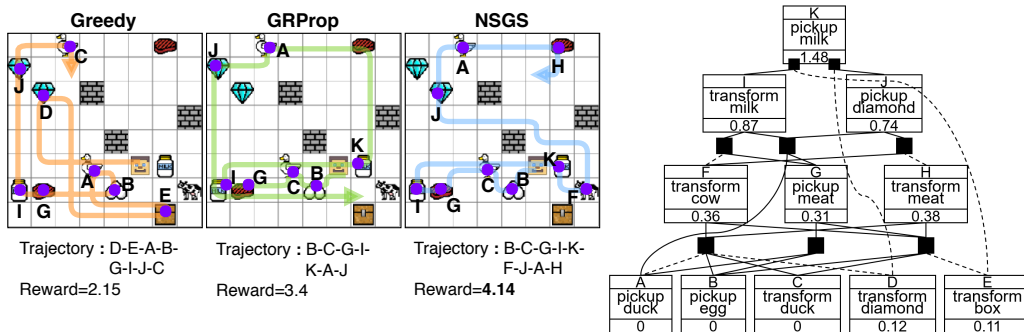

Figure 6: Example trajectories of Greedy, GRProp, and NSGS agents given 45 steps on Playground domain. The subtask graph includes NOT operation and distractor (subtask D, E, and H). We removed stochasticity in environment for the controlled experiment. Greedy agent executes the distractors since they give positive immediate rewards, which makes it impossible to execute the subtask K which gives the largest reward. GRProp and NSGS agents avoid distractors and successfully execute subtask K by satisfying its preconditions. After executing subtask K, the NSGS agent found a shorter path to execute remaining subtasks than the GRProp agent and gets larger reward.

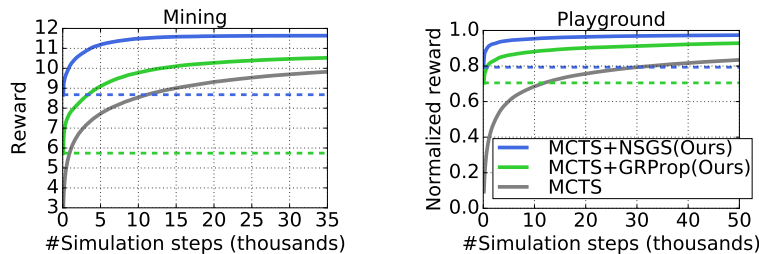

Figure 7: Performance of MCTS+NSGS, MCTS+GRProp and MCTS per the number of simulated steps on (Left) **Eval** of Mining domain and (Right) **D2** of Playground domain (see Table 1).

number of subtasks on Playground domain. Table 1 summarizes the results in terms of normalized reward $\bar{R} = (R - R_{min})/(R_{max} - R_{min})$ where $R_{min}$ and $R_{max}$ correspond to the average reward of the Random and the Optimal policy respectively. Due to large number of subtasks ($>16$) in Mining domain, the Optimal policy was intractable to be evaluated. Instead, we reported the un-normalized mean reward. Though the performance degrades as the subtask graph becomes larger as expected, NSGS generalizes well to larger subtask graphs and consistently outperforms all the other agents on Playground and Mining domains in zero-shot setting. In adaptation setting, NSGS performs slightly better than zero-shot setting by fine-tuning on the subtask graphs in evaluation set. Independent agent learned a policy comparable to Greedy, but performs much worse than NSGS.

## 5.4 Qualitative Result

Figure 5 visualizes trajectories of agents on Mining domain. Greedy policy mostly focuses on subtasks with immediate rewards (e.g., get string, make bow) that are sub-optimal in the long run. In contrast, NSGS and GRProp agents focus on executing subtask H (make stone pickaxe) in order to collect materials much faster in the long run. Compared to GRProp, NSGS learns to consider observation also and avoids subtasks with high cost (e.g., get coal).

Figure 6 visualizes trajectories on Playground domain. In this graph, there are distractors (e.g., D, E, and H) and the reward is delayed. In the beginning, Greedy chooses to execute distractors, since they gives positive reward while subtasks A, B, and C do not. However, GRProp observes non-zero gradient for subtasks A, B, and C that are propagated from the parent nodes. Thus, even though the reward is delayed, GRProp can figure out which subtask to execute. NSGS learns to understand long-term dependencies from GRProp, and finds shorter path by also considering the observation.

## 5.5 Combining NSGS with Monte-Carlo Tree Search

We further investigated how well our NSGS agent performs compared to conventional search-based methods and how our NSGS agent can be combined with search-based methods to further improve the performance. We implemented the following methods (see Appendix for the detail):

- MCTS: An MCTS algorithm with UCB [31] criterion for choosing actions.
- MCTS+NSGS: An MCTS algorithm combined with our NSGS agent. NSGS policy was used as a rollout policy to explore reasonably good states during tree search, which is similar to AlphaGo [32].
- MCTS+GRProp: An MCTS algorithm combined with our GRProp agent similar to MCTS+NSGS.

The results are shown in Figure 7. It turns out that our NSGS performs as well as MCTS method with approximately 32K simulations on Playground and 11K simulations on Mining domain, while GRProp performs as well as MCTS with approximately 11K simulations on Playground and 1K simulations on Mining domain. This indicates that our NSGS agent implicitly performs long-term reasoning that is not easily achievable by a sophisticated MCTS, even though NSGS does not use any simulation and has never seen such subtask graphs during training. More interestingly, MCTS+NSGS and MCTS+GRProp significantly outperforms MCTS, and MCTS+NSGS achieves approximately $0.97$ normalized reward with 33K simulations on Playground domain. We found that the Optimal policy, which corresponds to normalized reward of $1.0$, uses approximately 648M simulations on Playground domain. Thus, MCTS+NSGS performs almost as well as the Optimal policy with only $0.005\%$ simulations compared to the Optimal policy. This result implies that NSGS can also be used to improve simulation-based planning methods by effectively reducing the search space.

## 6 Conclusion

We introduced the subtask graph execution problem which is an effective and principled framework of describing complex tasks. To address the difficulty of dealing with complex subtask dependencies, we proposed a graph reward propagation policy derived from a differentiable form of subtask graph, which plays an important role in pre-training our neural subtask graph solver architecture. The empirical results showed that our agent can deal with long-term dependencies between subtasks and generalize well to unseen subtask graphs. In addition, we showed that our agent can be used to effectively reduce the search space of MCTS so that the agent can find a near-optimal solution with a small number of simulations. In this paper, we assumed that the subtask graph (e.g., subtask dependencies and rewards) is given to the agent. However, it will be very interesting future work to investigate how to extend to more challenging scenarios where the subtask graph is unknown (or partially known) and thus need to be estimated through experience.

## Acknowledgments

This work was supported mainly by the ICT R&D program of MSIP/IITP (2016-0-00563: Research on Adaptive Machine Learning Technology Development for Intelligent Autonomous Digital Companion) and partially by DARPA Explainable AI (XAI) program #313498 and Sloan Research Fellowship.

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
