[Supplementary Material]

# Supplementary Material:
# Hierarchical Reinforcement Learning for Zero-shot Generalization with Subtask Dependencies

## 1 Details of the Task

We define each task as an MDP tuple $\mathcal{M}_G = (\mathcal{S}, \mathcal{A}, \mathcal{P}_G, \mathcal{R}_G, \rho_G, \gamma)$ where $\mathcal{S}$ is a set of states, $\mathcal{A}$ is a set of actions, $\mathcal{P}_G : \mathcal{S} \times \mathcal{A} \times \mathcal{S} \to [0,1]$ is a task-specific state transition function, $\mathcal{R}_G : \mathcal{S} \times \mathcal{A} \to \mathbb{R}$ is a task-specific reward function and $\rho_G : \mathcal{S} \to [0,1]$ is a task-specific initial distribution over states. We describe the subtask graph $G$ and each component of MDP in the following paragraphs.

**Subtask and Subtask Graph**  The subtask graph consists of $N$ subtasks that is a subset of $\mathcal{O}$, the subtask reward $\mathbf{r} \in \mathbb{R}^N$, and the precondition of each subtask. The set of subtasks is $\mathcal{O} = \mathcal{A}_{int} \times \mathcal{X}$, where $\mathcal{A}_{int}$ is a set of primitive actions to interact with objects, and $\mathcal{X}$ is a set of all types of interactive objects in the domain. To execute a subtask $(a_{int}, obj) \in \mathcal{A}_{int} \times \mathcal{X}$, the agent should move on to the target object $obj$ and take the primitive action $a_{int}$.

**State**  The state $\mathbf{s}_t$ consists of the observation $\mathbf{obs}_t \in \{0,1\}^{W \times H \times C}$, the completion vector $\mathbf{x}_t \in \{0,1\}^N$, the time budget $step_t$ and the eligibility vector $\mathbf{e}_t \in \{0,1\}^N$. An observation $\mathbf{obs}_t$ is represented as $H \times W \times C$ tensor, where $H$ and $W$ are the height and width of map respectively, and $C$ is the number of object types in the domain. The $(h, w, c)$-th element of observation tensor is 1 if there is an object $c$ in $(h, w)$ on the map, and 0 otherwise. The time budget indicates the number of remaining time-steps until the episode termination. The completion vector and eligibility vector provides additional information about $N$ subtasks. The details of completion vector and eligibility vector will be explained in the following paragraph.

**State Distribution and Transition Function**  Given the current state $(\mathbf{obs}_t, \mathbf{x}_t, \mathbf{e}_t)$, the next step state $(\mathbf{obs}_{t+1}, \mathbf{x}_{t+1}, \mathbf{e}_{t+1})$ is computed from the subtask graph $G$. In the beginning of episode, the initial time budget $step_t$ is sampled from a pre-specified range $N_{step}$ for each subtask graph (See section 10 for detail), the completion vector $\mathbf{x}_t$ is initialized to a zero vector in the beginning of the episode $\mathbf{x}_0 = [0, \ldots, 0]$ and the observation $\mathbf{obs}_0$ is sampled from the task-specific initial state distribution $\rho_G$. Specifically, the observation is generated by randomly placing the agent and the $N$ objects corresponding to the $N$ subtasks defined in the subtask graph $G$. When the agent executes subtask $i$, the $i$-th element of completion vector is updated by the following update rule:

$$x_{t+1}^i = \begin{cases} 1 & \text{if} \quad e_t^i = 1 \\ x_t^i & \text{otherwise} \end{cases}. \tag{1}$$

The observation is updated such that agent moves on to the target object, and perform corresnponding primitive action (See Section 9 for the full list of subtasks and corresponding primitive actions on Mining and Playground domain). The eligibility vector $\mathbf{e}_{t+1}$ is computed from the completion vector $\mathbf{x}_{t+1}$ and subtask graph $G$ as follows:

$$e_{t+1}^i = \underset{j \in Child_i}{\text{OR}} \left( y_{AND}^j \right), \tag{2}$$

$$y_{AND}^i = \underset{j \in Child_i}{\text{AND}} \left( \widehat{x}_{t+1}^{i,j} \right), \tag{3}$$

$$\widehat{x}_{t+1}^{i,j} = x_{t+1}^j w^{i,j} + (1 - x_{t+1}^j)(1 - w^{i,j}), \tag{4}$$

where $w^{i,j} = 0$ if there is a `NOT` connection between $i$-th node and $j$-th node, otherwise $w^{i,j} = 1$. Intuitively, $\widehat{x}_t^{i,j} = 1$ when $j$-th node does not violate the precondition of $i$-th node. Executing each subtask costs different amount of time depending on the map configuration. Specifically, the time cost is given as the Manhattan distance between agent location and target object location in the grid-world plus one more step for performing a primitive action.

**Task-specific Reward Function**   The reward function is defined in terms of the subtask reward vector $\mathbf{r}$ and the eligibility vector $\mathbf{e}_t$, where the subtask reward vector $\mathbf{r}$ is the component of subtask graph $G$ the and eligibility vector is computed from the completion vector $\mathbf{x}_t$ and subtask graph $G$ as Eq. 4. Specifically, when agent executes subtask $i$, the reward given to agent at time step $t$ is given as follows:

$$r_t = \begin{cases} r^i & \text{if} \quad e_t^i = 1 \\ 0 & \text{otherwise} \end{cases} . \tag{5}$$

## 2   Experiment on Hierarchical Task Network

We compared with our methods with the recent graph-based multitask RL works [1–3]. However, these methods cannot be applied to our problem for two main reasons: 1) they aim to solve a single-goal task, which means they can only solve a subset of our problem, and 2) they require search or learning during test time, which means they cannot be applied in zero-shot generalization setting. Specifically, each trajectory in single-goal task is assumed to be labeled as success or failure depending on whether the goal was achieved or not, which is necessary for these methods [1–3] to infer the task structure (e.g., hierarchical task network (HTN) [4]). Since our task setting is more general and not limited to a single goal task, the task structure with multiple goals cannot be inferred with these methods.

For a direct comparison, we simplified our problem into single-goal task as follows. 1) We set a single goal; set all the subtask reward to 0, except the top-level subtask, and set it as terminal state. 2) We removed the cost, time budget, and observation, and set $\gamma = 1$. After constructing the task network such as HTN, these methods [1–3] execute task by planning [1] or learning a policy [2, 3] during test stage. Accordingly, we evaluated HTN-plan method [1] in planning setting, and allowed learning in test time for [2, 3]. Note that these methods cannot execute a task in zero-shot setting, while our NSGS can do it by learning an embedding of subtask graph; it is the main reason why our method performs much better than these methods in the following two experiments.

| Adaptation (HTN) | |
| --- | --- |
| Method | $R$ |
| NSGS (Ours) | **.90** |
| HTN-Independent | .31 |

Figure 1: Planning performance of MCTS+NSGS, MCTS+GRProp and HTN-Plan on HTN subtask graph in Playground domain.

Table 1: Adaptation performance (normalized reward) of NSGS and HTN-Independent on HTN subtask graph in Playground domain.

### 2.1   Comparison with HTN-Planning

Hayes and Scassellati [1] performed planning on the inferred task network to find the optimal solution. Thus, we implemented **HTN-Plan** with MCTS as in section 5.5, and compared with ours in planning setting. We evaluated our **MCTS+NSGS** and **MCTS+GRProp** for comparison. The figure shows that our **MCTS+NSGS** and **MCTS+GRProp** agents outperform HTN-Plan by a large margin.

### 2.2   Comparison with HTN-based Agent

Instead of planning, Ghazanfari and Taylor [2] learned an hierarchical RL (HRL) agent on the constructed HTN during testing. Thus, we evaluated it in adaptation setting (i.e., learning during test time). To this end, we implemented an HRL agent, HTN-Independent, which is a policy over option trained on each subtask graph independently, similar to Independent agent (see section 5.2). The result shows that our NSGS agent can find the solution much faster than HTN-Independent agent due to zero-shot generalization ability.

Huang et al. [3] inferred the subtask graph from the visual demonstration in testing. Since the environment state is available in our setting, providing demonstration amounts to providing the solution. Thus we couldn't compare with it.

# 3 Details of NSGS Architecture

Figure 2: An example of R3NN construction for a given subtask graph input. The four encoders $(b_{\theta_a}, b_{\theta_o}, t_{\theta_a},$ and $t_{\theta_o})$ are cloned and connected according to the input subtask graph where the cloned models share the weight. For simplicity, only the output embeddings of bottom-up and top-down OR encoder were specified in the figure.

**Task module**   Figure 2 illustrates the structure of the task module of NSGS architecture for a given input subtask graph. Specifically, the task module was implemented with four encoders: $b_{\theta_a}, b_{\theta_o}, t_{\theta_a},$ and $t_{\theta_o}$. The input and output of each encoder is defined in the main text section **4.1** as:

$$\phi_{bot,o}^i = b_{\theta_o}\left(x_t^i, e_t^i, step, \sum_{j \in Child_i} \phi_{bot,a}^j\right), \qquad \phi_{bot,a}^j = b_{\theta_a}\left(\sum_{k \in Child_j}\left[\phi_{bot,o}^k, w_+^{j,k}\right]\right), \tag{6}$$

$$\phi_{top,o}^i = t_{\theta_o}\left(\phi_{bot,o}^i, r^i, \sum_{j \in Parent_i}\left[\phi_{top,a}^j, w_+^{i,j}\right]\right), \quad \phi_{top,a}^j = t_{\theta_a}\left(\phi_{bot,a}^j, \sum_{k \in Parent_j} \phi_{top,o}^k\right), \tag{7}$$

For bottom-up process, the encoder takes the output embeddings of its children encoders as input. Similarly, for top-down process, the encoder takes the output embeddings of its parent encoders as input. The input embeddings are aggregated by taking element-wise summation. For $\phi_{bot,a}^i$ and $\phi_{top,o}^i$, the embeddings are concatenated with $w_+^{i,j}$ to deal with NOT connection before taking the element-wise summation. Then, the summed embedding is concatenated with all additional input as defined in Eq. 6 and 7, which is further transformed with three fully-connected layers with 128 units. The last fully-connected layer outputs 128-dimensional output embedding. The embeddings are transformed to reward scores as via: $\mathbf{p}_t^{reward} = \mathbf{\Phi}_{top}^\top \mathbf{v}$, where $\mathbf{\Phi}_{top} = [\phi_{top,o}^1, \ldots, \phi_{top,o}^N] \in \mathbb{R}^{E \times N}$, $E$ is the dimension of the top-down embedding of OR node, and $\mathbf{v} \in \mathbb{R}^E$ is a weight vector for reward scoring. Similarly, the reward baseline is computed by $b_t^{reward} = \text{sum}(\mathbf{\Phi}_{top}^\top \tilde{\mathbf{v}})$, where $\text{sum}(\cdot)$ is the reduced-sum operation and $\tilde{\mathbf{v}}$ is the weight vector for reward baseline. We used parametric ReLU (PReLU) function as activation function.

**Observation module**   The network consists of BN1-Conv1(16x1x1-1/0)-BN2-Conv2(32x3x3-1/1)-BN3-Conv3(64x3x3-1/1)-BN4-Conv4(96x3x3-1/1)-BN5-Conv5(128x3x3-1/1)-BN6-Conv6(64x1x1-1/0)-FC(256).   The output embedding of FC(256) was then concatenated with the number of remaining time step $step_t$. Finally, the network has two fully-connected output layers for the cost score $\mathbf{p}_t^{cost} \in \mathbb{R}^N$ and the cost baseline $b_t^{cost} \in \mathbb{R}$. Then, the policy of NSGS is calculated by adding reward score and cost score, and taking softmax:

$$\pi(\mathbf{o}_t|\mathbf{s}_t, G) = \text{Softmax}(\mathbf{p}_t^{reward} + \mathbf{p}_t^{cost}). \tag{8}$$

The baseline output is obtained by adding reward baseline and cost baseline:

$$V_{\theta'}(\mathbf{s}_t, G) = b_t^{reward} + b_t^{cost}. \tag{9}$$

# 4 Details of Learning NSGS Agent

**Learning objectives**   The NSGS architecture is first trained through policy distillation and finetuned using actor-critic method with generalized advantage estimator. During policy distillation, the KL divergence between NSGS and teacher policy (GRProp) is minimized as follows:

$$\nabla_\theta \mathcal{L}_1 = \mathbb{E}_{G \sim \mathcal{G}_{train}} \left[ \mathbb{E}_{s \sim \pi_\theta^G} \left[ \nabla_\theta D_{KL} \left( \pi_T^G || \pi_\theta^G \right) \right] \right], \tag{10}$$

where $\theta$ is the parameter of NSGS architecture, $\pi_\theta^G$ is the simplified notation of NSGS policy with subtask graph input $G$, $\pi_T^G$ is the simplified notation of teacher (GRProp) policy with subtask graph input $G$, $D_{KL} \left( \pi_T^G || \pi_\theta^G \right) = \sum_a \pi_T^G \log \frac{\pi_T^G}{\pi_\theta^G}$ and $\mathcal{G}_{train} \subset \mathcal{G}$ is the training set of subtask graphs.

For both policy distillation and fine-tuning, we sampled one subtask graph for each 16 parallel workers, and each worker in turn sample a mini-batch of 16 world configurations (maps). Then, NSGS generates total 256 episodes in parallel. After generating episode, the gradient from 256 episodes are collected and averaged, and then back-propagated to update the parameter. For policy distillation, we trained NSGS for 40 epochs where each epoch involves 100 times of update. Since our GRProp policy observes only the subtask graph, we only trained task module during policy distillation. The observation module was trained for auxiliary prediction task; observation module predicts the number of step taken by agent to execute each subtask.

After policy distillation, we finetune NSGS agent in an end-to-end manner using actor-critic method with generalized advantage estimation (GAE) [5] as follows:

$$\nabla_\theta \mathcal{L}_2 = \mathbb{E}_{G \sim \mathcal{G}_{train}} \left[ \mathbb{E}_{s \sim \pi_\theta^G} \left[ -\nabla_\theta \log \pi_\theta^G \sum_{l=0}^{\infty} \left( \prod_{n=0}^{l-1} (\gamma\lambda)^{k_n} \right) \delta_{t+l} \right] \right], \tag{11}$$

$$\delta_t = r_t + \gamma^{k_t} V_{\theta'}^\pi(\mathbf{s}_{t+1}, G) - V_{\theta'}^\pi(\mathbf{s}_t, G), \tag{12}$$

where $k_t$ is the duration of option $\mathbf{o}_t$, $\gamma$ is a discount factor, $\lambda \in [0, 1]$ is a weight for balancing between bias and variance of the advantage estimation, and $V_{\theta'}^\pi$ is the critic network parameterized by $\theta'$. During training, we update the critic network to minimize $\mathbb{E} \left[ (R_t - V_{\theta'}^\pi(\mathbf{s}_t, G))^2 \right]$, where $R_t$ is the discounted cumulative reward at time $t$.

**Hyperparameters**   For both finetuning and policy distillation, we used RMSProp optimizer with the smoothing parameter of 0.97 and epsilon of 1e-6. When distilling agent with teacher policy, we used learning rate=1e-4 and multiplied it by 0.97 on every epoch for both Mining and Playground domain. For finetuning, we used learning rate=2.5e-6 for Playground domain, and 2e-7 for Mining domain. For actor-critic training for NSGS, we used $\alpha = 0.03$, $\lambda = 0.96$, $\gamma = 0.99$.

# 5 Details of AND/OR Operation and Approximated AND/OR Operation

In section 4.2, the output of $i$-th AND and OR node in subtask graph were defined using AND and OR operation with multiple input. They can be represented in logical expression as below:

$$\underset{j \in Child_i}{\text{OR}} \left( y^j \right) = y^{j_1} \vee y^{j_2} \vee \ldots \vee y^{j_{|Child_i|}}, \tag{13}$$

$$\underset{j \in Child_i}{\text{AND}} \left( y^j \right) = y^{j_1} \wedge y^{j_2} \wedge \ldots \wedge y^{j_{|Child_i|}}, \tag{14}$$

where $j_1, \ldots, j_{|Child_i|}$ are the elements of a set $Child_i$ and $Child_i$ is the set of inputs coming from the children nodes of $i$-th node. Then, these AND and OR operations are smoothed as below:

$$\underset{j \in Child_i}{\widetilde{\text{OR}}} \left( \widetilde{y}_{AND}^j \right) = h_{or} \left( \sum_{j \in Child_i} \widetilde{y}_{AND}^j \right), \tag{15}$$

$$\underset{j \in Child_i}{\widetilde{\text{AND}}} \left( \widehat{x}_t^{i,j} \right) = h_{and} \left( \sum_{j \in Child_i} \widehat{x}_t^{i,j} - |Child_i| + 0.5 \right), \tag{16}$$

where $h_{or}(x) = \alpha_o \tanh(x/\beta_o)$, $h_{and}(x) = \alpha_a \sigma(x/\beta_a)$, $\sigma(\cdot)$ is sigmoid function, and $\alpha_o, \beta_o, \alpha_a, \beta_a \in \mathbb{R}$ are hyperparameters to be set. We used $\beta_a = 0.6, \beta_o = 2, \alpha_a = 1/\sigma(0.25), \alpha_o = 1$ for Mining domain, and $\beta_a = 0.5, \beta_o = 1.5, \alpha_a = 1/\sigma(0.25), \alpha_o = 1$ for Playground domain.

# 6   Details of Subtask Executor

**Architecture**   The subtask executor has the same architecture of the parameterized skill archi­tecture of [6] with slightly different hyperparameters. The network consists of Conv1(32x3x3-1/1)-Conv2(32x3x3-1/1)-Conv3(32x1x1-1/0)-Conv4(32x3x3-1/1)-LSTM(256)-FC(256). The sub­task executor takes two task parameters ($q = [q^{(1)}, q^{(2)}]$) as additional input and computes $\chi(q) = \text{ReLU}(W^{(1)}q^{(1)} \odot W^{(2)}q^{(2)})$ to compute the subtask embedding, and further linearly transformed into the weights of Conv3 and the (factorized) weight of LSTM through multiplicative interaction as described above. Finally, the network has three fully-connected output layers for actions, termination probability, and baseline, respectively.

**Learning objective**   The subtask executor is trained through policy distillation and then finetuned. Similar to [6], we first trained 16 teacher policy network for each subtask. The teacher policy network consists of Conv1(16x3x3-1/1)-BN1(16)-Conv2(16x3x3-1/1)-BN2(16)-Conv3(16x3x3-1/1)-BN3(16)-LSTM(128)-FC(128). Similar to subtask executor network, the teacher policy network has three fully-connected output layers for actions, termination probability, and baseline, respectively. Then, the learned teacher policy networks are used as teacher policy for policy distillation to train subtask executor. During policy distillation, we train agent to minimize the following objective function:

$$\nabla_\xi \mathcal{L}_{1,sub} = \mathbb{E}_{\mathbf{o} \sim \mathcal{O}} \left[ \mathbb{E}_{s \sim \pi_\xi^{\mathbf{o}}} \left[ \nabla_\xi \left\{ D_{KL}\left(\pi_T^{\mathbf{o}} || \pi_\xi^{\mathbf{o}}\right) + \alpha L_{term} \right\} \right] \right], \tag{17}$$

where $\xi$ is the parameter of subtask executor network, $\pi_\xi^{\mathbf{o}}$ is the simplified notation of subtask executor given input subtask $\mathbf{o}$, $\pi_T^{\mathbf{o}}$ is the simplified notation of teacher policy for subtask $\mathbf{o}$, $L_{term} = -\mathbb{E}_{\mathbf{s}_t \in \tau_{\mathbf{o}}} \left[ \log \beta_\xi(\mathbf{s}_t, \mathbf{o}) \right]$ is the cross entropy loss of predicting termination, $\tau_{\mathbf{o}}$ is a set of state in which the subtask $\mathbf{o}$ is terminated, $\beta_\xi(s_t, \mathbf{o})$ is the termination probability output, and $D_{KL}\left(\pi_T^{\mathbf{o}} || \pi_\xi^{\mathbf{o}}\right) = \sum_a \pi_T^{\mathbf{o}}(a|s) \log \frac{\pi_T^{\mathbf{o}}(a|s)}{\pi_\xi^{\mathbf{o}}(a|s)}$. After policy distillation, we finetuned subtask executor using actor-critic method with generalized advantage estimation (GAE):

$$\nabla_\xi \mathcal{L}_{2,sub} = \mathbb{E}_{\mathbf{o} \sim \mathcal{O}} \left[ \mathbb{E}_{s \sim \pi_\xi^{\mathbf{o}}} \left[ -\nabla_\xi \log \pi_\xi \left( \mathbf{a}_t | \mathbf{obs}_t, \mathbf{o} \right) \sum_{k=0}^{\infty} (\gamma\lambda)^k \delta_{t+k} + \alpha \nabla_\xi L_{term} \right] \right], \tag{18}$$

where $\gamma \in [0, 1]$ is a discount factor, $\lambda \in [0, 1]$ is a weight for balancing between bias and variance of the advantage estimation, and $\delta_t = r_t + \gamma V^\pi(\mathbf{obs}_{t+1}; \xi') - V^\pi(\mathbf{obs}_t; \xi')$. We used $\lambda = 0.96$, $\gamma = 0.99$ for fine-tuning, and $\alpha = 0.1$ for both policy distillation and fine-tuning.

# 7   Details of LSTM Baseline

**Architecture**   The LSTM baseline consists of LSTM on top of CNN. The architecture of CNN is the same as the CNN architecture of observation module of NSGS described in the section 3, and the architecture of LSTM is the same as the LSTM architecture used in subtask executor described in the section 6. Specifically, it consists of BN1-Conv1(16x1x1-1/0)-BN2-Conv2(32x3x3-1/1)-BN3-Conv3(64x3x3-1/1)-BN4-Conv4(96x3x3-1/1)-BN5-Conv5(128x3x3-1/1)-BN6-Conv6(64x1x1-1/0)-LSTM(256)-FC(256). The CNN takes the observation tensor as an input and outputs an embedding. The embedding is then concatenated with other input vectors including subtask completion indicator $\mathbf{x}_t$, eligibility vector $\mathbf{e}_t$, and the remaining step $step_t$. Finally, LSTM takes the concatenated vector as an input and output the softmax policy with the parameter $\theta'$: $\pi_{\theta'}(\mathbf{o}_t | \mathbf{obs}_t, \mathbf{x}_t, \mathbf{e}_t, step_t)$.

**Learning objective**   The LSTM baseline was trained using actor-critic method. For the baseline, we found that the moving average of return works much better than learning a critic network, and used it for experiment. This is due to the characteristic of adaptation setting; in adaptation setting, the subtask graph is fixed and the agent is trained for only a small number of episodes such that the critic network is usually under-fitted. Similar to NSGS, the learning objective is given as

$$\nabla_{\theta'} \mathcal{L}_{LSTM} = \mathbb{E}_{s \sim \pi_{\theta'}^G} \left[ -\nabla_{\theta'} \log \pi_{\theta'} \left( \mathbf{o}_t | \mathbf{s}_t \right) \sum_{l=0}^{\infty} \left( \prod_{n=0}^{l-1} (\gamma\lambda)^{k_n} \right) \delta_{t+l} \right], \tag{19}$$

where $\gamma \in [0, 1]$ is a discount factor, $\lambda \in [0, 1]$ is a weight for balancing between bias and variance of the advantage estimation, $\delta_t = r_t + \gamma^{k_t} \overline{V}(t+1) - \overline{V}(t)$, and $\overline{V}(t)$ is the moving average of return at time step $t$. We used $\lambda = 0.96$ and $\gamma = 0.99$.

# 8 Details of Search Algorithms

Each iteration of Monte-Carlo tree search method consists of four stages: selection, expansion, rollout, and back-propagation.

- Selection: We used UCB criterion [7]. Specifically, the option for which the score below has the highest value is chosen for selection:

$$\text{score} = \frac{R_i}{n_i} + C_{UCB}\sqrt{\frac{\ln N}{n_i}}, \tag{20}$$

  where $R_i$ is the accumulated return at $i$-th node, $n_i$ is the number of visit of $i$-th node, $C_{UCB}$ is the exploration-exploitation balancing weight, and $N$ is the number of total iterations so far. We found that $C_{UCB} = 2\sqrt{2}$ gives the best result and used it for MCTS, MCTS+GRProp and MCTS+NSGS methods.

- Expansion: MCTS randomly chooses the remaining eligible subtask, while the subtask is chosen by NSGS policy for MCTS+NSGS method and GRProp policy for MTS+GRProp method. More specifically, MCTS+NSGS and MCTS+GRProp greedily chooses among the remaining subtasks based on NSGS and GRProp policy, respectively. Due to the memory limit, the expansion of search tree was truncated at the depth of 7 for Playground and 10 for Mining domains, and performed rollout after the maximum depth.

- Rollout: MCTS randomly executes an eligible subtask, while MCTS+NSGS and MCTS+GRProp execute the subtask with the highest probability given by NSGS and GRProp policies, respectively.

- Back-propagation: Once the episode is terminated, the result is back-propagated; the accumulated return $R_i$ and the visit count $n_i$ are updated for the nodes in the tree that agent visited within the episode, and the number of total iteration is updated as $N \leftarrow N + 1$.

# 9 Details of Environment

## 9.1 Mining

There are 15 types of objects: *Mountain*, *Water*, *Work space*, *Furnace*, *Tree*, *Stone*, *Grass*, *Pig*, *Coal*, *Iron*, *Silver*, *Gold*, *Diamond*, *Jeweler's shop*, and *Lumber shop*. The agent can take 10 primitive actions: *up*, *down*, *left*, *right*, *pickup*, *use1*, *use2*, *use3*, *use4*, *use5* and agent cannot moves on to the *Mountain* and *Water* cell. *Pickup* removes the object under the agent, and *use*'s do not change the observation. There are 26 subtasks in the Mining domain:

- Get wood/stone/string/pork/coal/iron/silver/gold/diamond: The agent should go to *Tree*/*Stone*/*Grass*/*Pig*/*Coal*/*Iron*/*Silver*/*Gold*/*Diamond* respectively, and take *pickup* action.

- Make firewood/stick/arrow/bow: The agent should go to *Lumber shop* and take *use1*/*use2*/*use3*/*use4* action respectively.

- Light furnace: The agent should go to *Furnace* and take *use1* action.

- Smelt iron/silver/gold: The agent should go to *Furnace* and take *use2*/*use3*/*use4* action respectively.

- Make stone-pickaxe/iron-pickaxe/silverware/goldware/bracelet: The agent should go to *Work space* and take *use1*/*use2*/*use3*/*use4*/*use5* action respectively.

- Make earrings/ring/necklace: The agent should go to *Jeweler's shop* and take *use1*/*use2*/*use3* action respectively.

The icons used in Mining domain were downloaded from `www.icons8.com` and `www.flaticon.com`. The *Diamond* and *Furnace* icons were made by Freepik from `www.flaticon.com`.

## 9.2 Playground

There are 10 types of objects: *Cow*, *Milk*, *Duck*, *Egg*, *Diamond*, *Heart*, *Box*, *Meat*, *Block*, and *Ice*. The *Cow* and *Duck* move by 1 pixel in random direction with the probability of 0.1 and 0.2, respectively. The agent can take 6 primitive actions: *up*, *down*, *left*, *right*, *pickup*, *transform* and agent cannot moves on to the *block* cell. *Pickup* removes the object under the agent, and *transform* changes the object under the agent to *Ice*. The subtask graph was randomly generated without any hand-coded template (see Section 10 for details).

## 10 Details of Subtask Graph Generation

### 10.1 Mining Domain

Figure 3: The entire graph of Mining domain. Based on this graph, we generated 640 subtask graphs by removing the subtask node that has no parent node.

The precondition of each subtask in Mining domain was defined as Figure 3. Based on this graph, we generated all possible sub-graphs of it by removing the subtask node that has no parent node, while always keeping subtasks A, B, D, E, F, G, H, I, K, L. The reward of each subtask was randomly scaled by a factor of $0.8 \sim 1.2$.

### 10.2 Playground Domain

| | | |
|---|---|---|
| **Nodes** | $N_T$ | number of tasks in each layer |
| | $N_D$ | number of distractors in each layer |
| | $N_A$ | number of AND node in each layer |
| | $r$ | reward of subtasks in each layer |
| **Edges** | $N_{ac}^+$ | number of children of AND node in each layer |
| | $N_{ac}^-$ | number of children of AND node with NOT connection in each layer |
| | $N_{dp}$ | number of parents with NOT connection of distractors in each layer |
| | $N_{oc}$ | number of children of OR node in each layer |
| **Episode** | $N_{step}$ | number of step given for each episode |

Table 2: Parameters for generating task including subtask graph parameter and episode length.

For training and test sample generation, the subtask graph structure was defined in terms of the parameters in table 2. To cover wide range of subtask graphs, we randomly sampled the parameters $N_A, N_O, N_{ac}^+, N_{ac}^-, N_{dc}$, and $N_{oc}$ from the range specified in the table 3 and 5, while $N_T$ and $N_D$ was manually set. We prevented the graph from including the duplicated AND nodes with the same children node(s). We carefully set the range of each parameter such that at least 500 different subtask graphs can be generated with the given parameter ranges. The table 3 summarizes parameters used to generate training and evaluation subtask graphs for the Playground domain.

| | | |
|---|---|---|
| Train (=**D1**) | $N_T$ | {6,4,2,1} |
| | $N_D$ | {2,1,0,0} |
| | $N_A$ | {3,3,2}-{5,4,2} |
| | $N_{ac}^+$ | {1,1,1}-{3,3,3} |
| | $N_{ac}^-$ | {0,0,0}-{2,2,1} |
| | $N_{dp}$ | {0,0,0}-{3,3,0} |
| | $N_{oc}$ | {1,1,1}-{2,2,2} |
| | $r$ | {0.1,0.3,0.7,1.8}-{0.2,0.4,0.9,2.0} |
| | $N_{step}$ | 48-72 |
| **D2** | $N_T$ | {7,5,2,1} |
| | $N_D$ | {2,2,0,0} |
| | $N_A$ | {4,3,2}-{5,4,2} |
| | $N_{ac}^+$ | {1,1,1}-{3,3,3} |
| | $N_{ac}^-$ | {0,0,0}-{2,2,1} |
| | $N_{dp}$ | {0,0,0,0}-{3,3,0,0} |
| | $N_{oc}$ | {1,1,1}-{2,2,2} |
| | $r$ | {0.1,0.3,0.7,1.8}-{0.2,0.4,0.9,2.0} |
| | $N_{step}$ | 52-78 |
| **D3** | $N_T$ | {5,4,4,2,1} |
| | $N_D$ | {1,1,1,0,0} |
| | $N_A$ | {3,3,3,2}-{5,4,4,2} |
| | $N_{ac}^+$ | {1,1,1,1}-{3,3,3,3} |
| | $N_{ac}^-$ | {0,0,0,0}-{2,2,1,1} |
| | $N_{dp}$ | {0,0,0,0,0}-{3,3,3,0,0} |
| | $N_{oc}$ | {1,1,1,1}-{2,2,2,2} |
| | $r$ | {0.1,0.3,0.6,1.0,2.0}-{0.2,0.4,0.7,1.2,2.2} |
| | $N_{step}$ | 56-84 |
| **D4** | $N_T$ | {4,3,3,3,2,1} |
| | $N_D$ | {0,0,0,0,0,0} |
| | $N_A$ | {3,3,3,3,2}-{5,4,4,4,2} |
| | $N_{ac}^+$ | {1,1,1,1,1}-{3,3,3,3,3} |
| | $N_{ac}^-$ | {0,0,0,0,0}-{2,2,1,1,0} |
| | $N_{dp}$ | {0,0,0,0,0,0}-{0,0,0,0,0,0} |
| | $N_{oc}$ | {1,1,1,1,1}-{2,2,2,2,2} |
| | $r$ | {0.1,0.3,0.6,1.0,1.4,2.4}-{0.2,0.4,0.7,1.2,1.6,2.6} |
| | $N_{step}$ | 56-84 |

Table 3: Subtask graph parameters for training set and tasks **D1**∼**D4**.

| Zero-Shot Performance | | | | | |
|---|---|---|---|---|---|
| | Playground($R$) | | | | Mining($R$) |
| Task | **D1** | **D2** | **D3** | **D4** | **Eval** |
| NSGS (Ours) | **.820** | **.785** | **.715** | **.527** | **8.19** |
| NSGS-task (Ours) | .773 | .730 | .645 | .387 | 6.51 |
| GRProp (Ours) | .721 | .682 | .623 | .424 | 6.16 |
| NSGS-scratch (Ours) | .046 | .056 | .062 | .106 | 3.68 |
| Random | 0 | 0 | 0 | 0 | 2.79 |

Table 4: Zero-shot generalization performance on Playground and Mining domain. NSGS-scratch agent performs much worse than NSGS and GRProp agent on Playground and Mining domain.

# 11 Ablation Study on Neural Subtask Graph Solver Agent

## 11.1 Learning without Pre-training

We implemented **NSGS-scratch** agent that is trained with actor-critic method from scratch without pre-training from GRProp policy to show that pre-training plays a crucial role for training our NSGS agent. Table 4 summarizes the result. NSGS-scratch performs much worse than NSGS, suggesting that pre-training is important in training NSGS. This is not surprising as our problem

is combinatorially intractable (e.g. searching over optimal sequence of subtasks given an unseen subtask graph).

## 11.2 Ablation Study on the Balance between Task and Observation Module

We implemented **NSGS-task** agent that uses only the task module without observation module to compare the contribution of task module and observation module of NSGS agent. Overall, our NSGS agent outperforms the NSGS-task agent, showing that the observation module improves the performance by a large margin.

## 12 Experiment Result on Subtask Graph Features

To investigate how agents deal with different types of subtask graph components, we evaluated all

Figure 4: Normalized performance on subtask graphs with different types of dependencies.

agents on the following types of subtask graphs:

- 'Base' set consists of subtask graphs with AND and OR operations, but without NOT operation.
- 'Base-OR' set removes all the OR operations from the base set.
- 'Base+Distractor' set adds several distractor subtasks to the base set.
- 'Base+NOT' set adds several NOT operations to the base set.
- 'Base+NegDistractor' set adds several negative distractor subtasks to the base set.
- 'Base+Delayed' set assigns zero reward to all subtasks but the top-layer subtask.

Note that we further divided the set of Distractor into Distractor and NegDistractor. The distractor subtask is a subtask without any parent node in the subtask graph. Executing this kind of subtask may give an immediate reward but is sub-optimal in the long run. The negative-distractor subtask is a subtask with only and at least one NOT connection to parent nodes in the subtask graph. Executing this subtask may give an immediate reward, but this would make other subtasks not executable. Table 5 summarizes the detailed parameters used for generating subtask graphs. The results are shown in Figure 4. Since 'Base' and 'Base-OR' sets do not contain NOT operation and every subtask gives a positive reward, the greedy baseline performs reasonably well compared to other sets of subtask graphs. It is also shown that the gap between NSGS and GRProp is relatively large in these two sets. This is because computing the optimal ordering between subtasks is more important in these kinds of subtask graphs. Since only NSGS can take into account the cost of each subtask from the observation, it can find a better sequence of subtasks more often.

In 'Base+Distractor', 'Base+NOT', and 'Base+NegDistractor' cases, it is more important for the agent to carefully find and execute subtasks that have a positive effect in the long run while avoiding distractors that are not helpful for executing future subtasks. In these tasks, the greedy baseline tends to execute distractors very often because it cannot consider the long-term effect of each subtask in principle. On the other hand, our GRProp can naturally screen out distractors by getting zero or negative gradient during reward back-propagation. Similarly, GRProp performs well on 'Base+Delayed' set because it gets non-zero gradients for all subtasks that are connected to the final rewarding subtask. Since our NSGS was distilled from GRProp, it can handle delayed reward or distractors as well as (or better than) GRProp.

|  |  |  |
|---|---|---|
| **Base** | $N_T$ | {4,3,2,1} |
| | $N_D$ | {0,0,0,0} |
| | $N_A$ | {3,3,2}-{4,3,3} |
| | $N_{ac}^+$ | {1,1,2}-{3,2,2} |
| | $N_{ac}^-$ | {0,0,0}-{0,0,0} |
| | $N_{dp}$ | {0,0,0,0}-{0,0,0,0} |
| | $N_{oc}$ | {1,1,1}-{2,2,2} |
| | $N_{step}$ | 40-60 |
| **-OR** | $N_{oc}$ | {1,1,1}-{1,1,1} |
| **+Distractor** | $N_D$ | {2,1,0,0} |
| **+NOT** | $N_{ac}^+$ | {0,0,0}-{3,2,2} |
| **+NegDistractor** | $N_D$ | {2,1,0,0} |
| | $N_{dp}$ | {0,0,0,0}-{3,3,0,0} |
| **+Delayed** | $r$ | {0,0,0,1.6}-{0,0,0,1.8} |

Table 5: Subtask graph parameters for analysis of subtask graph components.