[Reviews · NeurIPS 2018]

Reviewer 1



In this paper, the author(s) introduced a new problem called subtask graph execution, where the agent is required to execute a given subtask graph in an optimal way. There are indeed some innovations in this work. Such as, neural subtask graph solver (NSS) which encodes the subtask graph using a recursive neural network, and to overcome the difficulty of training, the author(s) proposed a novel non-parametric gradient-based policy to pre-train the NSS agent and further fine-tune it through actor-critic method. The effectiveness is verified in two environments. 1.This paper is well written and easy to read, but there are a few misspelled words and symbol indicates error in this paper, such as, the ‘subtaks’in line 145 and the symbol of the top-down embedding line 147. 2.Two main contributions are presented for solving the proposed problem and training effectively. 3.The author(s) provide sufficient experimental evidence that the RProp and the NSS further improves the performance through fine-tuning with actor-critic method and performs much better than Greedy in zero-shot and adaption setting.

Reviewer 2



In this paper, the authors studied the problem of subtask graph execution, that aims to find an optimal way to execute the given subtask graph. To address this problem, a deep RL framework is proposed to explore the subtask graph using R3NN. Empirical studies are performed on two domains with sets of subtask graphs. The paper is well-written and easy to follow. However, my main concerns are that the application-driven motivation for the studied problem is not clear to me. The authors may want to provide more discussion to answer (1) why the studied problem is vital to our real life, i.e., the subtask graph may be hard to obtain, and (2) why the proposed methods are advanced than the existing graph-based multi-task learning tasks. Besides, more experiments should be provided by comparing with more state-of-the-art graph-based multi-task learning approaches. Strengths: (1) a novel problem of subtask graph execution. (2) the paper is well-written and easy to follow. (3) interesting empirical studies on two real domains with sets of subtask graphs Weakness: (1) the application-driven motivation of the studied problem is not clear. (2) more comparison experiments should be conducted by including recent graph-based multi-task learning methods.

Reviewer 3



The paper introduces an RL problem where the agent is required to execute a given subtask graph which describes a set of subtasks and their dependency, and proposes a neural subtask graph solver (NSS) to solve this problem. In NSS, there are an observation module to capture the environment information using CNN, and a task module to encode the subtask graph using recursive-reverse-recursive neural network (R3NN). A non-parametric reward-propagation policy (RProp) is proposed to pre-train the NSS agent and further finetune it through actor-critic method. Pros: 1. In general, the problem introduced in this paper is interesting and the method which uses CNN to capture the observation information and R3NN to encode the subtask graph is a good idea. 2. The method is evaluated in two experiments which outperforms the baselines in some different settings. Cons: 1. Writing: many details of the proposed method are included in the supplementary material which makes it difficult to understand by reading the main paper only. For example, even the objective is not defined in the main paper and all the training processes are presented in the supplementary material. 2. The baseline used in this paper is simple. Why not compare to the schemes listed in the related work section?